# Bilirubin Metabolism Does Not Influence Serum Bile Acid Profiles According to LC–MS: A Human Case-Control Study

**DOI:** 10.3390/ijms26062475

**Published:** 2025-03-10

**Authors:** Tamara Christina Stelzer, Ralf Krüger, Paola Gloria Ferrario, Christine Mölzer, Marlies Wallner, Rodrig Marculescu, Daniel Doberer, Andrew Cameron Bulmer, Karl-Heinz Wagner

**Affiliations:** 1Department of Nutritional Sciences, University of Vienna, 1090 Vienna, Austria; 2Vienna Doctoral School for Pharmaceutical, Nutritional and Sport Sciences (PhaNuSpo), University of Vienna, 1090 Vienna, Austria; 3Department of Physiology and Biochemistry of Nutrition, Max Rubner-Institut, Federal Research Institute of Nutrition and Food, 76131 Karlsruhe, Germany; 4Department of General Surgery, Division of Visceral Surgery, Medical University of Vienna, Vienna General Hospital, Währinger Gürtel 18-20, 1090 Vienna, Austria; 5Institute of Dietetics and Nutrition, University of Applied Sciences FH JOANNEUM, 8010 Graz, Austria; 6Department of Laboratory Medicine, Medical University of Vienna, 1090 Vienna, Austria; 7Vienna Health Care Group, Department of Respiratory and Critical Care Medicine, Klinik Floridsdorf, Bruenner Straße 68, 1210 Vienna, Austria; 8School of Pharmacy and Medical Sciences, Griffith University, Southport, QLD 4215, Australia

**Keywords:** bilirubin, bile acids, hyperbilirubinemia

## Abstract

In addition to their role in lipid absorption, bile acids (BAs) are also known for several relevant (endocrine) activities including contributing to the regulation of energy homeostasis and some non-communicable diseases (NCDs). Furthermore, moderately elevated unconjugated bilirubin levels, as observed in Gilbert’s syndrome (GS), may protect against NCDs. We therefore hypothesized that the BA profile in GS subjects differs from that of normo-bilirubinemic individuals. To test this, we performed a human case-control study, in which GS (*n* = 60) and controls (*n* = 60) were matched for age and gender, and serum BA concentrations were measured by liquid-chromatography mass spectrometry (LC–MS). Despite analyzing a comprehensive panel of BAs, no significant differences between the two groups were observed. These data suggest that bile acid concentrations are similar between groups, indicating that altered bilirubin metabolism unlikely influences their transport into the blood.

## 1. Introduction

Bile acids (BAs) play an important role in lipid metabolism by forming micelles to enhance the intestinal absorption of fat-soluble compounds [1]. Recently, BAs were shown to have multiple (endocrine) functions by binding to both nuclear as well as cytoplasmic receptors, such as the farnesol X receptor (FXR), TGR5 (Takeda G protein-coupled receptor 5), or the vitamin D receptor (VDR) [2,3]. Through FXR activation, they are involved in glucose metabolism by stimulating glycogen synthesis [4]. Furthermore, BAs reduce triglyceride levels by suppressing lipogenesis and hepatic secretion [5]. A more recent concept is the potential involvement of BAs in the control of energy metabolism [6]. This concept is based on the observation that mice fed a diet enriched in cholic acid (CA) had increased energy expenditure in brown fat and were therefore protected from dietary fat-induced obesity and insulin resistance [5,6,7]. Therefore, BAs are also potentially implicated in the pathogenesis of non-communicable diseases (NCDs), such as cardiovascular disease (CVD) and type 2 diabetes mellitus (T2DM). Van de Peppel et al. recently described that BAs have potential in the treatment of gastrointestinal, hepatic, and metabolic disorders due to their influence on different physiological pathways [1].

Moderately elevated unconjugated bilirubin (UCB) levels, as observed in Gilbert’s syndrome (GS), may protect against NCDs. GS individuals have lower cholesterol levels, reduced body mass index (BMI), and lower blood glucose levels [8]. In addition to contributing to energy homeostasis, BAs are also involved in glucose metabolism and in the depletion of the cholesterol pool, as BA production requires approximately 500 mg of cholesterol per day, making them one of the major cholesterol “consumers” in the human body [9]. Therefore, it could be assumed that GS individuals might have a different BA composition compared to the population with physiological UCB levels. Furthermore, Vidimce and colleagues described a different bile flow and biliary bile acid secretion in female hyperbilirubinemic Gunn rats compared to littermates [10]. This suggests that elevated bilirubin levels may influence bile acid metabolism and/or excretion, which in turn could affect circulating concentrations. To test this hypothesis, we investigated whether individuals with mildly elevated bilirubin levels had a different serum BA composition compared to healthy controls, in order to gain more insights into different mechanisms behind the protective role of the GS state.

## 2. Methods

The BiliHealth study, an observational case-control study, was conducted between June 2014 and January 2015. Initially, 128 individuals aged 20 to 74 years with and without GS were recruited from the general Austrian population by direct advertising (bulletin boards, posters, and flyers) and from the department’s subject database. Of these individuals, eight were excluded after the initial screening due to medical and lifestyle factors (supplement use, medication, elevated liver enzymes, smoking). Detailed exclusion criteria have been described elsewhere [11,12,13]. Finally, 60 individuals with GS and 60 age and gender matched healthy controls completed the study. Individuals were assigned to the GS or control group (C) based on their fasting UCB levels measured by high pressure liquid chromatography (HPLC) (</≥17.1 µM, the cutoff level for GS). Regarding gender distribution, 80 men and 40 women participated in the study. Furthermore, participants were also divided into two age groups (<; ≥35 years): 53 participants were over the age of 35 and 67 under the age of 35. The study was approved by the Ethics Committee of the Medical University of Vienna (#1164/2014) and was conducted according to the guidelines of the Declaration of Helsinki. Signed written informed consent was obtained from each participant included in the study.

In an initial screening, liver enzymes, such as gamma-glutamyltransferase (γ-GT), alanine aminotransferase (ALT), aspartate aminotransferase (AST), and hemolysis parameters, were assessed by standard clinical chemistry techniques. All individuals were required to fast on the day before participation and had to follow a 400 kcal/d fasting protocol of 16 ± 1 h before blood sampling.

Anthropometric measurements were taken as described elsewhere [13]. In addition, blood glucose, glycated hemoglobin (HbA1c), insulin, and C-peptide were analyzed in plasma by the laboratory of the General Hospital of Vienna, Austria, using routine methods.

Dietary habits were assessed using food frequency questionnaires, as described by Moelzer et al. [11]. The following three categories of foods were analyzed: “health foods”, “red meat”, and “snack food”. Additionally, alcohol consumption was assessed. Values are given in times of consumption per week.

BA concentrations were analyzed using LC–MS/MS (Xevo TQD + Acquity H-Class; Waters, Eschborn, Germany), following the method of Frommherz et al. [14] adapted to UPLC conditions. BAs were extracted from blood plasma by solid phase extraction (SPE) using polymeric RP cartridges (Strata-X 33, 30 mg/1 mL; Phenomenex, Aschaffenburg, Germany). Purified BAs were separated on a RP C18 column (HSS T3, 100 × 2.1 mm, 1.8 µM; Waters, Eschborn, Germany) using an 0.4 mL/min acetonitrile gradient at pH 5.1 (ammonium acetate). Individual BAs were detected in negative electrospray ionization (ESI) and multiple reaction monitoring (MRM)/pseudo-MRM mode. Calibrators and control samples were prepared by stripping blood plasma with activated charcoal and spiking with an appropriate amount of BA mix. Quantification was achieved by internal matrix-matched calibration using deuterated BA analogs. Matrix samples were spiked before and after extraction and compared to matrix-free standards, showing moderate matrix effects and >86% recovery for all BA species. In total, 16 different BAs were identified (see Table 1).

Statistical analysis was performed with IBM SPSS (version 29). Bile acid concentrations are displayed as median (25th percentile; 75th percentile), and in nmol/L, if not otherwise specified. Data were tested for normality of distribution using the Shapiro–Wilk test. As not all data were normally distributed, the non-parametric Mann–Whitney U-test was used for analysis. Principal component analysis (PCA) was used to identify specific BA profiles. Independently, further statistical analysis was performed, using R (version 4.0.0). Specifically, the statistical workflow described in [15] was applied. Here, UCB was the continuous variable of interest and its association with the BAs was investigated, using the mathematical models and the R-packages (version 4.0.0) described in [15]. Statistically significant differences were considered at *p* < 0.05.

## 3. Results

A significant difference in BMI existed between GS (mean ± stdv; 22.85 ± 3.03) and controls (C) (25.23 ± 4.86; *p* < 0.001). There was no difference in BMI between young GS (22.64 ± 3.06) and C (23.14 ± 3.25). However, in participants older than 35 years, GS had significantly reduced BMI (GS > 35: 22.93 ± 2.88; C > 35: 27.97 ± 5.26; *p* = 0.002). Lean body mass (LBM), as a percentage of body weight, was significantly higher in young participants (78.29 ± 5.77%) compared to those over 35 years of age (74.36 ± 9.20%; *p* = 0.003), with significant differences between GS and C. The effect of having GS was greater in older participants (GS: 77.99 ± 7.59; C: 70.89 ± 9.42; *p* = 0.003) than in younger individuals (GS: 77.66 ± 6.14; C: 78.80 ± 5.47; *p* = 0.019). The same was true for body fat (subjects >35 years: GS: 22.01 ± 7.57%; C: 29.11 ± 9.42%; *p* = 0.003; subjects ≤35: GS: 22.34 ± 6.14%; C: 21.20 ± 5.47%; *p* = 0.019).

Hepatic parameters (AST, ALT, γ-GT, and lactate dehydrogenase) were not significantly different between groups (all *p* > 0.05). Regarding food consumption, no difference was observed in red meet intake (*p* = 0.450), healthy food (*p* = 0.263), snack food (*p* = 0.863), and alcohol (*p* = 0.235). Details to these values are provided elsewhere [8].

GS individuals had significantly lower glucose levels (82.49 mg/dL ± 10.56) than C subjects (85.78 mg/dL ± 8.14; *p* = 0.001). In addition, insulin (GS: 5.07 ± 2.84; C: 7.80 ± 6.14; *p* < 0.001) and C-peptide levels (GS: 1.40 ± 0.60; C: 1.81 ± 0.99; *p* = 0.004) were also significantly reduced in the GS group. HbA1c was not significantly different (*p* = 0.116).

In addition to these previous study results, blood BAs were quantified in order to test the hypothesis of an altered BA profile related to the elevated bilirubin levels characteristic of the GS phenotype. Therefore, 16 different BAs were analyzed by UPLC–MS/MS and are shown in Table 1. There were no differences between GS and controls in any of the BAs measured. Furthermore, there were no significant correlations between UCB and BA concentrations, as indicated in Table 1. Notably, we observed high biological variations, due to some extreme values in both groups, but particularly for the GS group.

However, PCA also failed to reveal any significant pattern. Overall, it was observed that GS participants had a slightly more heterogeneous BA profile than the control group, based on the PCA; however, it was not significant. Nevertheless, the overall BA distribution of the entire study population was very similar, as shown Figure 1.

**Figure 1 ijms-26-02475-f001:**
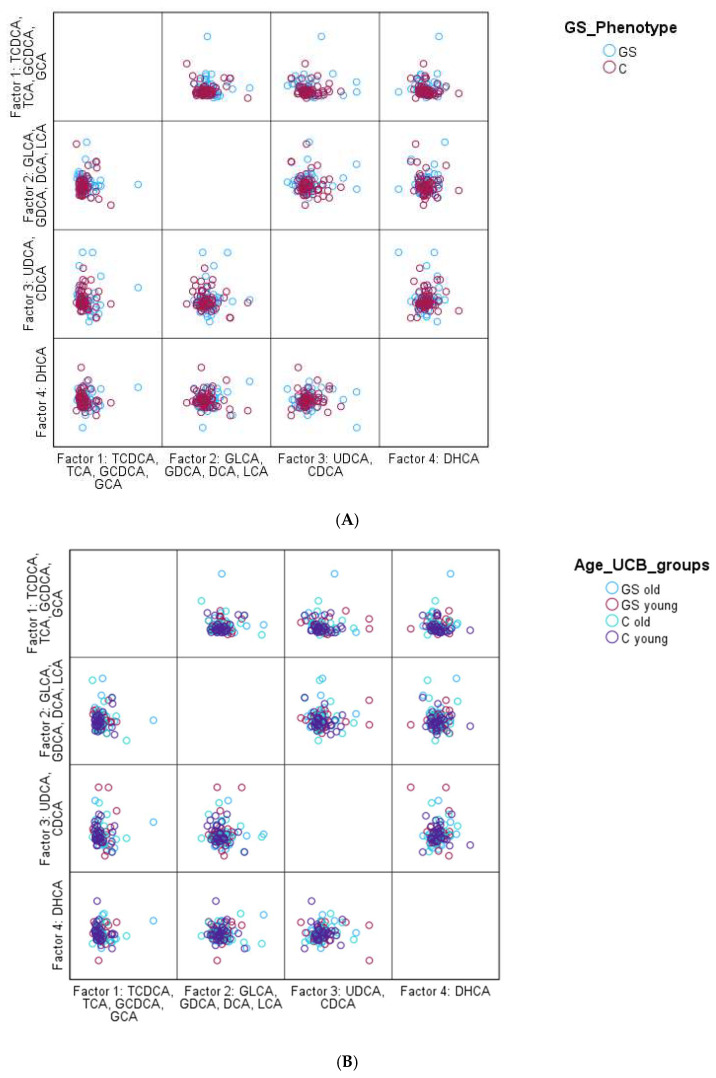
PCA for the identification of a specific BA profile (**A**):GS phenotype vs. controle; (**B**): GS phenotype of young vs. old: <; ≥35 years (**C**): GS phenotype of young vs. old: <; ≥35 years and gender: male vs. female). Factors were named according to the highest load of each BA. For factor 1, the highest loads were the highest for TCDCA, TCA, GCDCA, and GCA, for factor 2 it was GLCA, GDCA, DCA, and LCA, for factor 3 UDCA and CDCA, and for factor 4 it was DHCA.

**Table 1 ijms-26-02475-t001:** Bile acid concentrations are displayed in nmol/L as median (25th percentile and 75th percentile) per group (GS and control). *p*-values originate from Mann–Whitney U-Tests, no significant differences were observed between cases and controls. † In one sample BAs were not detectable due to technical issues. The following BAs were analyzed: CA (cholic acid), CDCA (chenodeoxycholic acid), deoxycholic acid (DCA), dehydrocholic acid (DHCA), glycocholic acid (GCA), glycochenoxycholic acid (GCDCA), glycodeoxycholic acid (GDCA), glycolithocholic acid (GLCA), glycoursodeoxycholic acid (GUDCA), lithocholic acid (LCA), taurocholic acid (TCA), taurochenodeoxycholic acid (TCDCA), taurodeoxycholic acid (TDCA), taurolithocholic acid (TLCA), tauroursodeoxycholic acid (TUDCA), and ursodeoxycholic acid (UDCA).

Bile Acid	Group	N	Median (P25; P75)	*p*-Value
CA	GS	59 †	20.0 (12.4; 41.6)	0.764
	C	60	22.3 (12.0; 52.9)	
CDCA	GS	59	53.9 (17.7; 143)	0.992
	C	60	56.0 (17.5; 246)	
DCA	GS	59	153 (88.5; 293)	0.552
	C	60	201 (93.3; 288)	
DHCA	GS	59	4.64 (2.04; 6.39)	0.886
	C	60	4.85 (2.45; 5.88)	
GCA	GS	59	102 (52.9; 220)	0.807
	C	60	113 (46.6; 238)	
GCDCA	GS	59	467 (289; 887)	0.726
	C	60	495 (287; 925)	
GDCA	GS	59	131 (71.2; 351)	0.730
	C	60	154 (76.8; 250)	
GLCA	GS	59	10.8 (6.47; 21.8)	0.467
	C	60	9.18 (4.91; 20.6)	
GUDCA	GS	59	51.5 (27.7; 133)	0.936
	C	60	55.7 (26.1; 101)	
LCA	GS	59	8.32 (4.63; 12.73)	0.617
	C	60	7.91 (4.43; 11.2)	
TCA	GS	59	14.4 (8.40; 34.1)	0.807
	C	60	14.9 (7.60; 27.5)	
TCDCA	GS	59	53.7 (25.8; 128)	0.262
	C	60	41.6 (25.0; 81.5)	
TDCA	GS	59	36.0 (14.7; 73.3)	0.189
	C	60	23.7 (12.1; 49.8)	
TLCA	GS	59	3.38 (1.95; 6.56)	0.453
	C	60	4.22 (2.09; 8.15)	
TUDCA	GS	59	2.71 (0.00; 8.24)	0.882
	C	60	2.61 (0.00; 5.19)	
UDCA	GS	59	11.2 (0.73; 32.4)	0.814
	C	60	7.45 (0.70; 49.6)	

Neither separation into two age groups (<; ≥35 years) nor into four age groups (20–27, 28–32, 33–46, 47–74 years) demonstrated significant differences in BA concentrations between groups.

Males (34.59 ± 49.66) had higher concentrations of UDCA than females (21.66 ± 43.23, *p* = 0.023). Concentrations of all other BAs were not significantly different between the sexes and when age, sex, and UCB levels were combined into one variable.

Furthermore, the absence of significant group-specific BA differences was confirmed by application of an independent statistical workflow (see Section 2).

## 4. Discussion

While GS individuals have reduced BMI, insulin, glucose, and C-peptide levels as well as a greater LBM compared to age and gender matched controls, no significant differences in serum BA concentrations were observed. Therefore, it appears that while bilirubin interferes with glucose and energy metabolism and affects body composition, it does so through mechanisms other than the BA profile.

Bile acids bind to FXR and other nuclear receptors. Through FXR binding, BAs are involved in glucose metabolism, where they activate FXR and insulin-dependent glucose uptake [4]. Furthermore, BA-activated FXR increases FGF15/19 (fibroblast growth factor), which stimulates hepatic glycogen synthesis [4]. Mice deficient inok FXR, for instance, develop insulin resistance [2]. Moreover, DCA activates the epidermal growth factor receptor (EGFR) in hepatocytes. EGFR is known to contribute to the progression of vascular dysfunction under diabetic conditions [2]. Since individuals with GS have significantly lower fasting glucose levels as well as a reduced risk of developing diabetes [16], it is reasonable to assume that DCA would be present at lower concentrations in GS than in controls. However, our analyses did not support this hypothesis. Therefore, it seems that mechanisms other than the BA profile lead to the reduced glucose concentrations and thus the lower risk of developing T2DM. We have demonstrated that a GS-specific phosphorylation state of adenosine monophosphate-activated protein kinase (AMPK) targets and proposes a potential “metformin-like” bilirubin effect [11]. Mechanistic studies [17] into bilirubin as peroxisome proliferator-activated receptor (PPAR) agonist have confirmed a direct interaction.

In addition to diabetes, BAs are also potentially implicated in the development of CVD. Elevated serum BAs are strongly associated with cardiac dysfunction. Some hydrophobic BAs, such as LCA, are involved in oxidative and inflammatory injury. Other more hydrophilic BAs, such as UDCA for instance, may exert preventive effects by acting as signaling molecules and activating FXR and TGR5. Therefore, the hydrophobicity of an individual’s serum BA pool may be of relevance. Moreover, an excess of serum BAs decreases fatty acid oxidation and may cause cardiac dysfunction [18]. Since the BA profile was not different in the GS group compared to controls, we conclude that the reduced CVD incidence in the GS phenotype is not related to the amount, hydrophobicity, or profile of the BA pool.

Bile acids contribute considerably to energy homeostasis and also play a role in the pathogenesis of metabolic diseases, such as overweight and obesity [6]. Bile acids could influence energy homeostasis via PPARα, but it is also possible that BAs could influence energy homeostasis via miRNAs [19]. Cholic acid, for instance, when added to the diet of mice, increased the energy expenditure of the animals and prevented obesity [19]. For that reason, it might have been expected that GS participants would have a higher concentration of CA, due to their significantly reduced BMI and body fat mass. However, this was not reflected in the results presented here, showing that energy homeostasis in GS subjects is not specifically regulated by a different BA profile, although there are data supporting the link between PPARα activation and bilirubin itself [20].

In humanized mice, it has been shown that the administration of BAs, such as UDCA, can successfully decrease bilirubin plasma levels, by either reducing bilirubin reabsorption or by enhancing fecal bilirubin excretion [3]. This fact points to the interaction between bilirubin and BA. In cholestatic conditions, it has been shown that BAs lower intrahepatic bilirubin levels, which shows an interaction between bilirubin and BAs [21]. Even though in all these observations it seems that BAs or the BA concentration is able to alter bilirubin levels, according to our results, bilirubin levels, on the other hand, seem to have no impact on BA concentrations in human serum. Vidimce et al. found that the concentration of several biliary BAs was significantly different between hyperbilirubinemic female Gunn rats and their normobilirubinemic littermates [10]. Therefore, it is possible that the effect of bilirubin on BAs in humans with mildly hyperbilirubinemic conditions is different to the one in the Gunn rat model, where the animals show much higher bilirubin concentrations.

Similar to prior work, there was no difference in the intake frequency of certain food groups between GS and controls, which proves that food intake had no influence on the BA profile.

## 5. Conclusions

Although we hypothesized that moderately elevated bilirubin levels and the complex GS condition would influence BA composition, our results indicate that the BA profile is independent of bilirubin levels and metabolism. Therefore, it is likely that mechanisms other than the BA profile are responsible for the protective health effect of the GS condition.

## Data Availability

The data that support the findings of this study are available from the corresponding author upon request.

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
