# Peer review of "Bilirubin Metabolism Does Not Influence Serum Bile Acid Profiles According to LC–MS: A Human Case-Control Study"

_ijms, 2025, doi:10.3390/ijms26062475_

Round 1
Reviewer 1 Report
Comments and Suggestions for Authors
Recommendations for authors
In the lines, 65-66, it is mentioned that: ”Initially 128 individuals aged 20 to 74 years with and without 65 GS were recruited from the general Austrian population.”. How were these patients recruited (from the general population)? How was the sample size calculated?
Median and interquartile range are usually/commonly reported as median (P25;P75), with the interquartile range as an interval. The use of the sign ± could be misleading.
For the Principal component analysis (PCA) how were the variables included (as they were or standardized)?
The discussion section could be enriched, particularly discussing the relationship between bilirubin metabolism and serum bile acid profile.
Author Response
Comment 1: In the lines, 65-66, it is mentioned that:” Initially 128 individuals aged 20 to 74 years with and without 65 GS were recruited from the general Austrian population.”. How were these patients recruited (from the general population)? How was the sample size calculated?
Response 1: We thank the reviewer for the comment. As described by Tosevska et al, 2016 sample size calculation for the “BiliHealth” study was performed based on data from a previous study on hyperbilirubinemia in humans (Wallner, 2013).
As described elsewhere (Mölzer et al, Zöhrer et al.), participants with and without GS were recruited between June 2014 and January 2015, by direct advertising (bulletin boards, posters and flyers) and from the department’s participant database. We added the according information in lines 66-67.
Comment 2: Median and interquartile range are usually/commonly reported as median (P25; P75), with the interquartile range as an interval. The use of the sign ± could be misleading.
Response 2: We agree with the reviewer’s comment and changed the ± sign and IQR to the (P25; P75) in the table of the manuscript, according to their advice.
Comment 3: The discussion section could be enriched, particularly discussing the relationship between bilirubin metabolism and serum bile acid profile.
Response 3: We appreciate the careful reading of our manuscript and enriched the discussion sections as advised in lines 358-373. "
In humanized mice it has been shown that the administration of BA, such as UDCA, can successfully decrease bilirubin plasma levels, by either reducing bilirubin reabsorption or by enhancing fecal bilirubin excretion (van der Schoor, 2021). This fact points to the interaction between bilirubin and BA. In cholestatic conditions it has been shown that BA lower intrahepatic bilirubin levels, which shows an interaction between bilirubin and BAs (Muchova, 2010). Even though, in all these observations it seems that BA or BA concentration is able to alter bilirubin levels, however, according to our results, bilirubin levels on the other hand seem to have no impact on BA concentrations, in human serum. Vidimce et al found that the concentration of several biliary BA was significantly different between hyperbilirubinemic female Gunn rats and their normobilirubinemic littermates (Vidimce et al, 2022). Therefore, it is possible that the effect of bilirubin on BAs in humans with mildly hyperbilirubnaemic conditions is different to the one in the Gunn rat model, where the animals show much higher bilirubin concentrations.
Similar to prior work there was no difference in the intake frequency of certain food groups between GS and controls, which proves that food intake had no influence of BA profile."
Reviewer 2 Report
Comments and Suggestions for Authors
The authors investigated the potential relationship between elevated unconjugated bilirubin (UCB) levels observed in Gilbert’s Syndrome and bile acid profiles in humans. I have the following concerns/suggestions.
- Some key regulators of BA metabolism were not addressed, such as dietary habits, comparison of liver enzyme activities between groups?
- The authors observed high biological variations, particularly in the GS group. Any impact on statistical analysis? GS participants had a slightly more heterogeneous BA profile. How was this quantified? Any subgroup analyses?
- LC-MS/MS, a sensitive and reliable method. However, several factors that might contribute to the results, such as, were any bile acids below the LOQ frequently, extraction efficiency for each bile acid, ion suppression, individual variability in circadian BA fluctuations?
- Additional bile acid species should be considered, such as those involved in glucuronidation, sulfation, intestinal microbiota metabolism.
- Negative findings are important, however, this could be due to limitations, the conclusion is not well supported by the presented results, and alternative explanations should be considered.
Author Response
Comment 1: Some key regulators of BA metabolism were not addressed, such as dietary habits, comparison of liver enzyme activities between groups?
Response 1: This an important point and we thank the reviewer for arising it. We added liver enzyme levels and dietary habits in both, GS and C subjects in the results section and state, that there are no significant differences between any of the parameters. Because of the limits for tables and figures in the “brief report” form, we had to add all the numbers in the text. Details were added in lines 123-126.
Comment 2: The authors observed high biological variations, particularly in the GS group. Any impact on statistical analysis?
Response 2: We thank the reviewer for the question. An impact of statistical analysis is rather unlikely for us since sample size was high enough to show significant differences for other parameter or parameter groups (Tosevska, 2016; Moelzer, 2017; Wallner, 2013). For the BAs the p-values were so high between groups, that there was even no indication or pattern visible for potential differences. This was also true for subgroup analysis.
Comment 3: GS participants had a slightly more heterogeneous BA profile. How was this quantified? Any subgroup analyses?
Response 3: We thank the reviewer for their input. We did analyze subgroups in the PCA, these analyses are shown in Figure 1b and 1c. However, since none of this analysis showed a significant effect, we did not analyze any further subgroups with lower n-numbers.
Comment 4: LC-MS/MS, a sensitive and reliable method. However, several factors that might contribute to the results, such as, were any bile acids below the LOQ frequently, extraction efficiency for each bile acid, ion suppression, individual variability in circadian BA fluctuations?
Response 4: We agree that analytical factors might contribute and have to be considered carefully. Method and validation were not described in full detail in the current short communication, partly due to word limits. We can answer your questions as follows:
- Indeed, we found a few BA species where several samples had values below the LOQ, mainly conjugated BA. However, the rate <LOQ was rather low, mostly below 10%. Moreover, the existence of missing values was thoroughly considered by both statistical workflows applied.
- We tested recovery and matrix effect simultaneously at two levels. Matrix samples were spiked before and after extraction and compared to matrix-free standards. Recovery was >86% for all BA species, and ion suppression was mostly below 30%. Furthermore, matrix suppression was accounted for by using appropriate internal standards. To provide basic validation details, we have added the following sentence in the manuscript:
“Matrix samples were spiked before and after extraction, and compared to matrix-free standards, showing moderate matrix effects and >86% recovery for all BA species.” Was added in lines 99-101.
- Variability in the circadian BA fluctuation is not method-related and is therefore not part of method validation. Circadian variance is a biological factor. However, the influence of time should be negligible in our study, since the timepoint of blood collection did not vary considerably.
Comment 4: Additional bile acid species should be considered, such as those involved in glucuronidation, sulfation, intestinal microbiota metabolism.
Response 4: Our method covers 16 abundant BA occurring in humans, including the important conjugates with glycine and taurine. A similar basic BA panel is used in many methods, further BA conjugates such as glucuronides or sulfates are often not covered. Especially sulfates tend to be labile and may require special precautions. We agree that those metabolites are interesting, but method extension and reanalysis of the whole study is currently not feasible and outside the scope of the present manuscript.
Comment 5: Negative findings are important, however, this could be due to limitations, the conclusion is not well supported by the presented results, and alternative explanations should be considered.
Response 5:
Although we hypothesized that moderately elevated bilirubin levels and the complex GS condition would influence BA composition, our results indicate clearly that the BA profile in humans is independent of bilirubin levels and metabolism. Therefore, it is not likely that the BA profile is a major player for already shown protective health effects of the GS condition. Further it seems that the more severe hyperbilirubinaemic condition in Gunn rats would support a link between bilirubin on BA, at least in biliary BA. Therefore, we can draw no clear conclusion from the data we have now and have to explore this further in future.
Some more lines in the discussion have been added - see answer to rev. 1
Reviewer 3 Report
Comments and Suggestions for Authors
Stelzer et al. reported their findings of serum bile acid profile in Gilbert’s syndrome (GS) patients. Based on the fact that GS patients have lower BMI, cholesterol, and blood glucose levels, which could be regulated by bile acids, the authors hypothesized that the bile acid profile in GS patients could be different from health people. However, LC-MS analysis of GS patients' and healthy people's serum indicated that there's no significant difference between the groups. The authors concluded that the metabolic syndromes in GS patients were not mediated by the change of bile acid profile. This report provides some valuable information about GS patient to people, although the results still need further verification with larger population. The hypothesis and method of this study were reasonable, and the results were well-discussed.
Author Response
Comment 1: Stelzer et al. reported their findings of serum bile acid profile in Gilbert’s syndrome (GS) patients. Based on the fact that GS patients have lower BMI, cholesterol, and blood glucose levels, which could be regulated by bile acids, the authors hypothesized that the bile acid profile in GS patients could be different from health people. However, LC-MS analysis of GS patients' and healthy people's serum indicated that there's no significant difference between the groups. The authors concluded that the metabolic syndromes in GS patients were not mediated by the change of bile acid profile. This report provides some valuable information about GS patient to people, although the results still need further verification with larger population. The hypothesis and method of this study were reasonable, and the results were well-discussed.
Response 1: We thank Reviewer 3 for carefully reading the manuscript and for the positive comments.
Round 2
Reviewer 1 Report
Comments and Suggestions for Authors
Dear authors
You did not respond to the question "For the Principal component analysis (PCA) how were the variables included (as they were or standardized)?" and what metric was used as an evaluation metric for PCA. Also it is not clear to me how the sample size was calculated?Author Response
Comment 1: You did not respond to the question "For the Principal component analysis (PCA) how were the variables included (as they were or standardized)?" and what metric was used as an evaluation metric for PCA.
Response 1: We deeply apologize for not having responded to that question in the last round, this was due to a mistake by copy-pasting the answers from our draft in word to the journal page here. Variables were not changed or standardized for analysis and were included as they were. The data underwent a regularizing data transformation. However, this did not change the outcome which is why we decided to include the data as it was. Our evaluation metric was the explained variance rotation.
Comment 2: Also it is not clear to me how the sample size was calculated?
Response 2: The estimation of the number of cases is based on the previous study from our group (Wallner et al. 2013). The calculation of this sample size was based on DNA damage, in the average of approx. 12 ± 2.5 % in the control group (non-GS) and a reduced DNA damage of 10 ± 2.5 % for the GS group with 10 ± 2.5 %. Assuming type I (0.05) and II errors (0.1), this resulted in in a case number of 33 persons per group. In this explorative-descriptive study, there was no primary outcome measure, however we aimed to increase the sample size by approx. 50 %.
This was the sample size calculation which was approved by statisticians and the Ethical Commission of the Medical University of Vienna.
Reviewer 2 Report
Comments and Suggestions for Authors
The authors made some improvements.
Author Response
Comment 1: The authors made some improvements.
Response 1: We thank the reviewer for the comment and the suggestions.
Round 3
Reviewer 1 Report
Comments and Suggestions for Authors
I have no other suggestions.
Author Response
Comment 1: I have no other suggestions.
Response 1: We thank the reviewer for carefully reading our manuscript and for all the suggestions before.